# Recent Insights into the Molecular Mechanisms of the Toll-like Receptor Response to Influenza Virus Infection

**DOI:** 10.3390/ijms25115909

**Published:** 2024-05-29

**Authors:** Mohammad Enamul Hoque Kayesh, Michinori Kohara, Kyoko Tsukiyama-Kohara

**Affiliations:** 1Department of Microbiology and Public Health, Faculty of Animal Science and Veterinary Medicine, Patuakhali Science and Technology University, Barishal 8210, Bangladesh; 2Department of Microbiology and Cell Biology, Tokyo Metropolitan Institute of Medical Science, Tokyo 156-8506, Japan; kohara-mc@igakuken.or.jp; 3Transboundary Animal Diseases Centre, Joint Faculty of Veterinary Medicine, Kagoshima University, Kagoshima 890-0065, Japan

**Keywords:** influenza virus, Toll-like receptor, TLR agonist, adjuvant

## Abstract

Influenza A viruses (IAVs) pose a significant global threat to human health. A tightly controlled host immune response is critical to avoid any detrimental effects of IAV infection. It is critical to investigate the association between the response of Toll-like receptors (TLRs) and influenza virus. Because TLRs may act as a double-edged sword, a balanced TLR response is critical for the overall benefit of the host. Consequently, a thorough understanding of the TLR response is essential for targeting TLRs as a novel therapeutic and prophylactic intervention. To date, a limited number of studies have assessed TLR and IAV interactions. Therefore, further research on TLR interactions in IAV infection should be conducted to determine their role in host–virus interactions in disease causation or clearance of the virus. Although influenza virus vaccines are available, they have limited efficacy, which should be enhanced to improve their efficacy. In this study, we discuss the current status of our understanding of the TLR response in IAV infection and the strategies adopted by IAVs to avoid TLR-mediated immune surveillance, which may help in devising new therapeutic or preventive strategies. Furthermore, recent advances in the use of TLR agonists as vaccine adjuvants to enhance influenza vaccine efficacy are discussed.

## 1. Introduction

Influenza viruses are emerging and re-emerging contagious pathogens that pose a significant threat to global health [1]. Influenza viruses belong to the *Orthomyxoviridae* family and are classified as types A, B, C, and the recently identified type D [2]. Influenza A viruses (IAVs) infect diverse host species, including birds, bats, pigs, and humans [3,4]. IAVs are enveloped viruses, containing a 13.5 kb genome composed of eight segments of negative-sense single-stranded RNA, encoding different proteins, including RNA-dependent RNA polymerase subunits (PB1, PB2, PA), viral glycoproteins hemagglutinin (HA) and neuraminidase (NA), viral nucleoprotein (NP), matrix protein M1 and M2, non-structural protein (NS1), and NS2 [5,6,7,8].

IAVs can be further characterized by their subtypes based on their surface glycoproteins, HA and NA [7]. HA, but not NA, is essential to initiate an infection, while the inhibition of NA has been reported to enhance infection [9,10]. HA and NA are antigenically distinct and are majorly targeted by influenza virus vaccines to produce neutralizing antibodies [11]. To date, 18 HA and 11 NA subtypes have been identified in IAVs, and a combination of these HA and NA subtypes can generate many different strains [4]. Vaccination remains the most effective intervention approach; however, mismatches between circulating and vaccine subtypes may reduce vaccine efficacy [12]. Moreover, multiple strategies adopted by IAVs can make it difficult to prevent successful infection and replication in the host [13]. 

The innate immune system, an important component of host immunity, confers protection against invading pathogens, including viruses [14]. The innate immune system detects conserved structures on microbes, such as pathogen-associated molecular patterns (PAMPs)/microbe-associated molecular patterns (MAMPs) and damage-associated molecular patterns (DAMPs), via the key innate immune-sensing receptors, germline-encoded pattern recognition receptors (PRRs) [15,16]. Different PRRs, such as Toll-like receptors (TLRs), retinoic acid-inducible gene I (RIG)-like receptors, nucleotide-binding oligomerization domain-containing protein-like receptors, C-type lectin receptors, AIM2-like receptors, and DNA-sensing receptors, are key innate immune components that recognize viral nucleic acids and proteins [16,17]. 

TLRs are important evolutionarily conserved innate immune components that play critical roles in viral infection [14,18]. TLRs are encoded by a large gene family, and a certain number of TLRs are present in different organisms: 10 TLRs (TLR1–TLR10) are found in humans and 12 TLRs (TLR1–TLR9 and TLR11–TLR13) are found in mice [19]. TLR1, TLR2, TLR4, TLR5, TLR6, and TLR10 are localized on the cell surface, while TLR3, TLR7, TLR8, and TLR9 are localized in the endoplasmic reticulum [20,21]. TLR1, TLR2, TLR4, TLR5, and TLR6 are involved in the detection of viral proteins [22], whereas TLR3, TLR7/TLR8, and TLR9 detect viral double-stranded RNA, single-stranded RNA (ssRNA), and CpG DNA, respectively [23,24,25,26]. However, the recognition of ssRNA by TLR7 and TLR8 is more complex, as long ssRNA is readily recognized by both TLR7 and TLR8 but the sequence-dependent activation of TLR7 and TLR8 occurs in the case of short RNA oligonucleotides [27]. 

TLRs contain a conserved N-terminal ectodomain of leucine-rich repeats, a single transmembrane domain, and a cytosolic Toll/interleukin (IL)-1 receptor (TIR) domain [14,28]. The TIR domain activates downstream signaling, and different TIR domain-containing cytosolic adaptor proteins, including myeloid differentiation factor 88 (MyD88), MyD88 adaptor-like (MAL or TIRAP), TIR-domain-containing adaptor protein-inducing IFN-β (TRIF or TICAM1), TRIF-related adaptor molecule (TRAM or TICAM2), and sterile α- and armadillo-motif-containing protein (SARM), are involved in the regulation of the TLR signaling pathways [29,30,31]. The MyD88 adaptor protein functions in almost all TLR signaling pathways, except in the TLR3 pathway [32]. TLR4 and TLR3 responses can activate the TRIF pathway, which finally activates IRF3; the TLR4 signaling pathway is unique and can activate both the MyD88 and TRIF signaling pathways [28]. TLRs remain the key players regulating viral infection dynamics [33,34,35,36,37,38] by inducing the production of interferons (IFNs), cytokines, and chemokines via several distinct signaling pathways, thus limiting infections and enhancing adaptive immune responses [16,39].

Despite the critical role of TLRs in the early recognition of pathogens and protection against infections, the TLR response may act as a double-edged sword; as a dysregulated TLR response, it may enhance immune-mediated pathology instead of protection [40,41,42,43,44,45]. Therefore, a clear understanding of the role of TLRs in any infection, including IAVs, is crucial for immunopathogenesis studies and also for the development of therapeutic and preventive interventions against IAV infection. Against this background, the focus of this article is on discussing recent progress in our understanding of the host TLR response to IAV infections and the mechanisms adopted by IAVs to avoid TLR-mediated immune surveillance, which may help devise new therapeutic or preventive strategies. Furthermore, recent advances in the use of TLR agonists as vaccine adjuvants for the production of influenza vaccines are discussed.

## 2. TLR Response to IAV Infection

It has been reported that the TLR2-specific ligand, peptidoglycan, reduces C5a surface receptor expression in neutrophils, whereas IAV increases the expression of the C5a receptor, indicating distinct effects on neutrophil surface receptor expression [46]. Both H1N1 and H3N2 IAV infections increase the surface expression of TLR2 in neutrophils [46]. A marked increase in TLR3 expression was reported upon IAV infection in human alveolar and bronchial epithelial cells, and TLR3 signaling has been reported to involve different signaling molecules, including mitogen-activated protein kinases, phosphatidylinositol 3-kinase, and TRIF [47]. 

It has been shown that pre-treatment of mice intranasally with TLR3 agonists, namely Poly ICLC and liposome-encapsulated Poly ICLC (LE Poly ICLC), provided protection against a lethal challenge of the avian H5N1 influenza strain (A/H5N1/chicken/Henan clade 2) as well as the seasonal influenza A/PR/8/34 (H1N1) and A/Aichi/2 (H3N2) strains [48], which is suggestive of the protective role of TLR3 in IAV infection. Pre-treatment of mice with a TLR9 agonist also protected them against a lethal challenge with H1N1 [48], indicating the protective role of TLR9 in IAV infection. 

Several viral proteins, such as the respiratory syncytial virus F protein, vesicular stomatitis virus glycoprotein, Ebola virus glycoprotein, and dengue virus non-structural protein 1, can activate TLR4 signaling and induce an inflammatory response. However, TLR4 activation in IAV infection has been attributed to host DAMPs, including the high-mobility group box 1 protein and oxidized phospholipids, which accumulate in response to infection rather than viral glycoproteins [49,50].

Increased expression of TLR3, TLR7, TLR8, and TLR9 was reported in monocytes and dendritic cells obtained from patients infected with IAV when compared with controls [51]. The association of increased TLR expression was observed with an increased cytokine response, including IL-6, sTNFR-1, CCL2/MCP-1, CXCL10/IP-10, and IFN-γ [51]. Using wild-type and TLR7-deficient mouse models of IAV infection, a previous study showed that the IFN-γ levels were significantly reduced in TLR7-deficient mice compared to wild-type mice, suggesting an important role of TLR7 in the induction of IFN-γ [52], which exhibits significant antiviral activity. Wang et al. reported that IAV could activate both TLR7 and TLR8 in human neutrophils, thereby inducing the production of inflammatory cytokines [53]. The role of TLR7 in the recognition of influenza virus and production of inflammatory cytokines has been demonstrated using murine neutrophils obtained from TLR7 knockout mice [53]. 

An increased expression of TLR2, TLR3, and TLR9 was observed in human patients infected with H1N1; however, no significant changes in TLR4, TLR7, or TLR8 expression were found [54]. An increased expression of IL-2, IL-6, IFN-γ, and TNF-α, and a decreased expression of IL-10, were observed in H1N1 infection [54]. Lee et al. reported the involvement of human TLR10 in recognizing influenza viral infection and showed that H1N1 and H5N1 IAVs are potent inducers of TLR10 expression [55].

Chicken TLR21, a functional homolog of mammalian TLR9, exhibits marked immunological responses to CpG oligodeoxynucleotides (ODNs), both in vitro and in vivo [56]. In ovo administration of CpG DNA in a chick embryo on day 18 and subsequent challenge with the H4N6 strain in the embryo on day 19 pre-hatching and day 1 post-hatching revealed reduced virus replication with enhanced NO production and macrophage recruitment in the lungs, suggesting a CpG DNA-mediated antiviral response, particularly against AIV infection in avian species [57].

The cotton rat (*Sigmodon hispidus*) is considered one of the reliable small animal models with which to study IAV infection [58]. An association between IFN-TLR expression and the pulmonary inflammatory response has been observed in lung tissues [59]. Treatment with the TLR3 agonist, Poly ICLC, inhibited replication of IAV in a dose-dependent manner, with 40 μg Poly ICLC displaying maximal and 1 μg Poly ICLC displaying minimal inhibition, suggesting the antiviral role of TLR3 response in this model [59]. A simplified overview of the response of TLRs to IAV infection is shown in Figure 1. Overall, TLR2–4, TRLR7, TLR8, and TLR9 are highly impacted during IAV infection, and the TLR3, TLR7 and TLR9 responses are important, playing an antiviral role in IAV infection.

## 3. Role of TLR Response in Pathology

IAV infection was reported to induce a decreased expression of TLR2 and TLR4, which might render individuals susceptible to bacterial infections [51]. A significant negative correlation between the TLR3/TLR8 expression levels and the viral load has been reported; a lower level of TLR expression was associated with a higher viral load, and a higher viral load was independently associated with the severity of diseases such as pneumonia and hypoxemia [51]. Dysregulated TLR expression has been observed in murine lung tissues of young adults and aged mice infected with mouse-adapted H1N1 or H3N2 strains [60].

An association between TLR3 signaling and IAV infection has been shown to increase the inflammatory response in a mouse model [61]. Surprisingly, a higher viral load in TLR3-deficient mice paradoxically showed a reduced lung pathology and a high survival rate of the mice compared to wild-type mice upon i.n. infection with 300 pfu of IAV [61]. Additionally, a marked expression of TLR3, IFN-γ, and tumor necrosis factor-alpha (TNF-α) was observed in the autopsy findings of lung tissues [62]. The TLR3 gene polymorphism rs5743313/CT, causing loss of function, might be linked to an increased risk of pneumonia in children during infection with A/H1N1/2009 influenza virus [63], and severe acute respiratory distress syndrome was reported in children with inherited TLR3 deficiency [64], suggesting a protective role of TLR3 in IAV infection. In humans, a missense mutation F303S in the TLR3 gene was shown to be associated with influenza-associated encephalopathy, a neurological symptom of severe influenza [65]. Overall, TLR3 signaling contributes antiviral effects against IAV infection in human studies; however, in mice models, it might render pathologic effects.

An association between the TLR4 response and acute lung injury has also been reported in mice [66]. However, the TLR7 response was reported to be favorable in a mouse model, where intranasal (i.n.) administration of imiquimod, a TLR7 agonist, showed an inhibition of peak viral replication, loss in body weight, and inflammation in the lungs and viral-induced lung dysfunction [67]. Sex-related variability in the cytokine response to IAV infection has been reported, where IL-10 production was significantly higher in the peripheral blood mononuclear cells of males compared to those of females upon stimulation of TLR8 with ssRNA40 [68].

Influenza viral protein, NP, interacts with TLR2, TLR4, and the NLR family pyrin domain-containing 3 (NLRP3) inflammasome, inducing the production of interleukin (IL)-1β and IL-6, which then leads to the induction of trypsin [69]. NP-induced trypsin enhances IAV infectivity and replication, thus playing a critical role in IAV-induced pathology [69].

## 4. Strategies for Curing IAV Infection by Targeting the TLR Signaling Pathway

The prior i.n. administration of the TLR2 agonist Pam2Cys showed protection against lethal influenza infection in mice; however, it had no effect on subsequent adaptive immune responses [70,71]. Another study showed that i.n. administration of a pegylated analog of the diacylated lipopeptide Pam2Cys, referred to as INNA-X, can activate TLR2 in the upper airways of mice, providing antiviral protection to the lungs [72]. In another study, in a mouse model, a single-stranded oligonucleotide (ssON)-mediated inhibition of TLR3 was reported to inhibit IAV infection both in vitro and in vivo [73].

TLR7 has been demonstrated to be a key component for triggering local and systemic activation of NK cells and IFN-gamma production in IAV infection, which are critical for providing an antiviral role [52]. IAV infection has been shown to induce an initial host defense response by activating TLR3 in the airway epithelium, resulting in increased ciliary activity and cilia-driven flow to enhance mucociliary clearance and the removal of viruses in a mouse model [74]. Intranasal administration of 3M-011, a synthetic TLR7/8 agonist, within 72 h to 6 h after viral inoculation was shown to induce the production of type I IFN and other cytokines that significantly inhibited H3N2 viral replication in a rat model [75].

## 5. Evasion of Immune Response by IAV Infection

Viruses have evolved by using multiple ways to modulate the host immune response, including the TLR response, and IAVs also utilize or adopt different strategies to avoid or inhibit host immune responses for successful infection and replication [76]. The HA protein of IAV facilitates ubiquitination and degradation of IFNAR, thereby reducing the type I IFN response [77]. IAVs employ the NA protein to prevent recognition of HA by natural cytotoxicity receptors and NKp46 and NKp44 receptors, resulting in reduced activity of NK cells toward virus elimination [78].

The anti-IFN activity of IAV NS1 is well-documented [79,80]. In addition to suppressing RIG-I signaling [81], NS1 also suppresses TLR3-mediated immune responses [82]. The NS1 protein of different IAV strains can perform various functions, and it is evident that NS1 can block IRF3 activation and IFN-β transcription [83]. 

In IAV infection, mature DCs showed the impairment of endogenous viral antigen presentation, thus affecting the adaptive immune response [84]. The host restriction factors SAM and HD domain-containing deoxynucleoside triphosphate triphosphohydrolase 1 (SAMHD1) has been shown to restrict IAV replication in A549 cells, suggesting that it can be used as a potential target to develop antivirals [85]. James et al. demonstrated that different viruses, including IAVs, can induce MAPK phosphatase 5 (MKP5) expression that can dephosphorylate IRF3, thereby suppressing the type I IFN response [86]. IAV H1N1 infection has been shown to upregulate the expression of PD-L1, leading to enhancement of viral replication and downregulation of the IFN response [87], thus implying a role of PD-L1 in IAV infection. It has been shown that SOX4 is activated during several viral infections, including IAV infection [88]. SOX4 negatively regulates TLR signaling, thus facilitating viral replication [88]. IAV infection leads to increased expression of SOCS1 and SOCS3, which inhibits the JAK/STAT pathway and disrupts the IFN-I- and IFN-II-mediated host defense [89,90,91]. In response to different H1N1, H3N2, H5N1, and H11N9 strains, SOCS5 levels are differentially regulated; they were found to be reduced in the primary epithelial cells of patients with COPD, leading to their increased susceptibility to influenza [92]. SOCS5 can restrict IAV infection in the airway epithelium by regulating the epidermal growth factor receptor, suggesting an association between SOCS5 and IAV infection [92]. Based on these findings, it is assumed that IAV and its proteins play an important role in inhibiting or suppressing the host innate immune response (Figure 2) by different mechanisms; therefore, a clear understanding of immune inhibition or evasion is essential for devising new therapeutic and preventive strategies to control IAV infection.

## 6. TLR Agonist as Influenza Vaccine Adjuvant

Poor immunogenicity hinders the development of effective peptide and DNA vaccines. Adjuvants are the substances that can be used as vaccine components to enhance vaccine efficacy. To overcome the poor immunogenicity of vaccine candidates, researchers are exploring novel adjuvants, including TLR agonists, which can provide both immunomodulatory and immunotherapeutic effects [93,94]. TLR ligands induce antigen-processing cells to stimulate different Th responses [95]. The use of TLR agonists as vaccine adjuvants has recently attracted increased attention on the part of vaccinologists and is being extensively studied [96]. However, the history of its use in immunotherapy dates back over a century, where bacteria or bacterial lysates were used for activation of the immune system [97,98,99]. William Coley, a pioneer of cancer immunotherapy, used bacterial lysates consisting of heat-inactivated *Streptococcus pyogenes* and *Serratia marcescens*, which are known as “Coley’s toxins”, for treating cancer [100]. Later, bacterial DNA was identified as the underlying component of the lysate that elicited the response [101].

TLR agonists as vaccine adjuvants show good profiles and promising responses to different vaccine candidates, including viral vaccines. CpG 1018, a TLR9 agonist, was recently approved to be used to boost the efficacy of hepatitis B VLP vaccine [102]. In a Phase III clinical trial, HEPLISAV-B, a recombinant HBV vaccine composed of HBsAg combined with the CpG 1018 adjuvant, produced a highly sustainable seroprotective response with fewer immunizations, including in individuals with poor vaccine response, while maintaining a favorable profile [103,104]. TLR agonists are potent immunomodulators capable of inducing the production of IFN, proinflammatory cytokines, and chemokines, and they have been shown to be promising candidates against many viral infections, including IAV [105,106]. 

Several types of influenza vaccines are currently in use, including inactivated whole-virus vaccine, split-virus vaccine, subunit vaccine, recombinant HA vaccine, and live attenuated influenza virus vaccine, which exhibit varied efficacies and overall efficacy of 70% [107,108,109]. The antigenic variations and genetic plasticity of the influenza viruses as well as the interference of pre-existing immunity render the vaccine efficacy low and unpredictable [110]. More effective influenza virus vaccines are needed to combat IAV infections, and the development of a universal influenza vaccine able to direct immunity toward conserved regions to cover a large spectrum of influenza strains remains a modern medical research goal [111]. Scientists are adopting multiple strategies, targeting the highly conserved epitopes of HA, NA, M2 extracellular domain and internal proteins of the influenza virus [110].

To combat future influenza outbreaks and/or overcome the poor efficacy of influenza vaccines, researchers have investigated the potential of TLR agonists to be used as adjuvants in the existing influenza vaccines or in the development of new vaccines. The TLR4 agonist glucopyranosyl lipid adjuvant–stable emulsion (GLA-SE) can enhance T cell responses when combined with influenza split-virus vaccines (SVVs), significantly improving vaccine-mediated protection against influenza in older adults [112]. In a Phase II clinical trial, GLA-SE enhanced the efficacy of the H5N1 plant-made virus-like particle vaccine, inducing a sustained polyfunctional and cross-reactive HA-specific CD4+ T cell response [113]. TLR4 (1Z105, a substituted pyrimido[5,4-b]indole specific for the TLR4-MD2 complex) and TLR7 ligands (1V270, a phospholipid-conjugated TLR7 agonist) combined with a recombinant HA antigen obtained from the A/Puerto Rico/8/1934 strain could induce rapid and cross-reactive humoral and cellular immunity, eliciting broad protection in murine models [114].

It has been shown that the combined use of monophosphoryl lipid A (MPL, a TLR4 agonist) and polyriboinosinic polyribocytidylic acid (poly I:C, a TLR3 agonist) with inactivated A/Puerto Rico/8/1934 (A/PR8) H1N1 influenza vaccine could enhance the protective efficacy against homologous influenza infection and induce significant memory T and B cell responses, minimizing illness outcomes in mouse models [115].

In a BALB/c mouse model, H1N1 influenza vaccine adjuvanted with a synthetic TLR7 agonist (imiquimod) showed higher efficacy against an early challenge with H1N1 compared to H1N1 influenza vaccine without an agonist [116], indicating the enhanced efficacy of the TLR7 agonist-adjuvanted vaccine. In another study, a TLR7/8 agonist (imidazoquinoline) combined with the H5N1 HA antigen was shown to broaden the antibody response in mice and ferrets and protect against homologous and heterologous virus challenges in ferrets [117], suggesting the increased efficacy of this influenza vaccine. Recombinant influenza HA protein adjuvanted with BBIQ, a pure TLR7 agonist, produced higher anti-influenza IgG1 and IgG2c responses in mice than in those immunized without an agonist [118]. In a Phase I/II clinical trial, the combination of a specific TLR3 agonist (rintatolimod) with FluMist was well tolerated, and its i.n. administration induced cross-reactive secretory IgA, which was effective against the H5N1, H7N9, and H7N3 strains [119].

Using non-human primate newborn models, several studies have shown the enhanced efficacy of a killed IAV vaccine adjuvanted either singly or in combination with flagellin (TLR5 agonist) and R848 (TLR7/8 agonist), which persistently induced the production of the total IAV-specific antibody [120,121,122]. In another study, Clemens et al. showed that HA adjuvanted with R848 could induce sustained production of IgG against the HA stem in a non-human primate newborn model and impact multiple cell types, including influenza-specific T follicular helper cells and Tregs with the potential to contribute to the HA stem response [123].

A broader immune response in BALB/c mice was reported upon vaccination with a recombinant influenza virus HA vaccine adjuvanted with TLR4 and TLR7 ligands [124]. In another study, administration of a licensed quadrivalent inactivated influenza vaccine adjuvanted with RIG-I (SDI-nanogel) and a TLR7/8 agonist (imidazoquinoline) enhanced the antibody and T cell responses and was correlated with protection against lethal influenza virus infection [125]. A recent study reported that the polymer-nanoparticle (PNP) hydrogel-based subcutaneous delivery of the HA protein along with a TLR7/8 adjuvant in a mouse model could increase the magnitude and duration of the antibody titers after a single injection [126], suggesting an effective vaccine delivery platform that enhances the efficacy of influenza subunit vaccines.

A recent study showed that mice vaccinated intranasally with a recombinant nucleoprotein (rNP) vaccine candidate adjuvanted with a TLR2/6 agonist, S-[2,3-bispalmitoyiloxy-(2R)-propyl]-R-cysteinyl-amido-monomethoxyl-poly-ethylene-glycol (BPPcysMPEG), exhibited markedly enhanced NP-specific IgG and IgG subclass titers than mice administered a vaccine without the adjuvant [127]. In addition, BPPcysMPEG improved the NP-specific cellular responses and induced a mixed Th1/Th2/Th17 immune profile in vaccinated mice [127]. Promising effects of TLR agonist-adjuvanted vaccines have already been reported in immunocompromised individuals against different viral infection [128].

Modifications of mRNA by replacing the nucleoside uridine with pseudouridine allow better stability, reduced immunogenicity, and higher translational capacity of the mRNA at the ribosomes [129,130]. The recent success of the COVID-19 mRNA vaccines reveals the potentiality in terms of the adaptability, universality, short production time, and ease in scaling up [131], which supports clinical trials for combatting other infections, including influenza (ClinicalTrials.gov ID: NCT05755620). In a Phase 1 clinical trial, mRNA vaccines against H10N8 and H7N9 were found to be well-tolerated and to elicit robust humoral immune responses [132]. A Phase 1/Phase 2 clinical study of multicomponent vaccine containing influenza and COVID-19 mRNAs is underway by ModernaTx, Inc. (ClinicalTrials.gov Identifier: NCT05827926). A Phase 1/2 randomized clinical trial of a quadrivalent, mRNA-based seasonal influenza vaccine (mRNA-1010) induced robust hemagglutination inhibition titers with no safety concerns [133], and to further evaluate the immunogenicity, reactogenicity and safety of the mRNA-1010 vaccine, a Phase 3 randomized controlled study is underway (ClinicalTrials.gov Identifier: NCT05827978). mRNA vaccines can show adjuvant effects and activate cellular RNA sensors that may upregulate RLRs, TLRs and C-type lectin receptors [134]. Different clinical trials are underway to develop universal influenza vaccines [135].

The increasing understanding of TLR agonists among vaccinologists has opened up a new toolbox [136]. The inclusion of TLR agonists in influenza vaccines holds promise for enhancing vaccine immunogenicity and efficacy [137]; however, the effects of TLR agonist adjuvants on long-term immune memory responses and their ability to improve immunity conferred by vaccines in different age groups need to be determined. Overall, TLR agonists hold promise for broadening the immune response that can be effective against strain mismatch and thus could enhance the efficacy of existing vaccines and new influenza vaccines under development to optimize current immunization approaches in the future. The TLR agonists that are under development to be used in influenza vaccines are shown in Table 1.

## 7. Conclusions

The association between TLR responses is critical during the investigation of influenza viruses. The TLR response in viral infections may act as a double-edged sword, and a balanced response is critical for the overall benefit of the host. A thorough understanding of the TLR response is essential for targeting TLRs as novel therapeutic and prophylactic interventions. TLR agonists also hold great promise as vaccine adjuvants for enhancing influenza vaccine efficacy; however, more studies are warranted to identify more suitable TLR agonists as vaccine adjuvants. Therefore, further research is essential to determine the exact role of TLRs in the pathogenesis and protection against IAV infections.

## Figures and Tables

**Figure 1 ijms-25-05909-f001:**
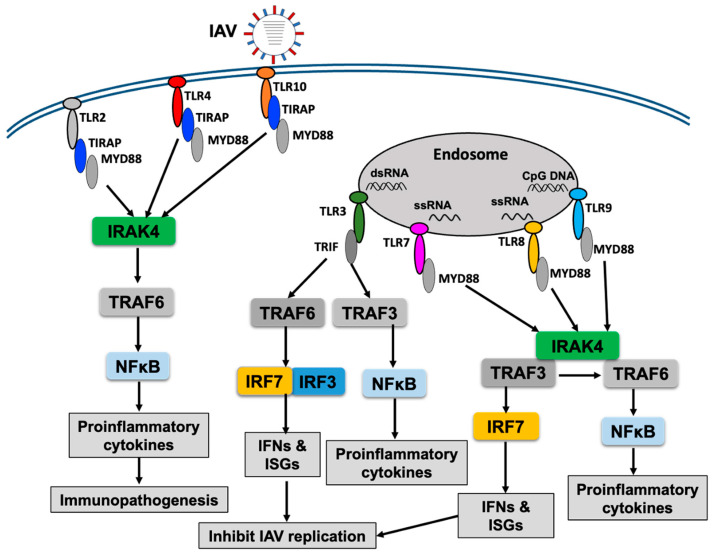
A simplified illustration of the TLR response to influenza A virus (IAV) infection. Upon recognition of viral protein or nucleic acids by the respective TLR, the downstream signaling pathways become activated, which finally culminates in the production of proinflammatory cytokines or chemokines and interferons, enhancing immunopathogenesis or inhibiting viral replication, respectively.

**Figure 2 ijms-25-05909-f002:**
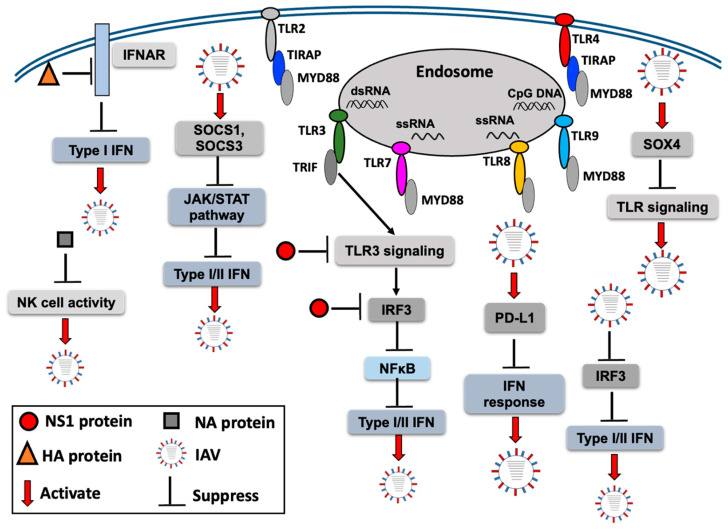
An overview of the host innate immune response inhibition by IAV and its proteins.

**Table 1 ijms-25-05909-t001:** TLR agonists as adjuvants for enhancing influenza vaccine efficacy.

Vaccine	TLR Agonist Adjuvant	Target TLR	Clinical Phase of Development	Effects on Host Immunity	ClinicalTrials.gov Identifier/Reference
FluMist vaccine	TLR3 agonist rintatolimod	TLR3	Phase I/II	Combination of rintatolimod and FluMist was well tolerated; enhanced cross-reactive secretory IgA	[119]
Split-virus influenza vaccine	Glucopyranosyl lipid adjuvant–stable emulsion (GLA-SE)	TLR4	-	Stimulated a Th1-cell response	[112]
Split-virus influenza vaccine	TLR4 agonist (1Z105) and TLR7 agonists (1V270)	TLR4 TLR7	-	Induced rapid, long-lasting, and balanced Th1- and Th2-type immunity in mice model	[114]
Inactivated H1N1 influenza vaccine	TLR7 agonists (1V270) TLR4 agonist 1Z105 (or its derivatives 2B182C)	TLR7 TLR4	-	Synergistically enhanced anti-HA and anti-NA IgG1 and IgG2a responses	[124]
H5N1 plant-made virus-like particle vaccine	GLA-SE	TLR4	Phase II	Induced both a humoral response and a sustained cross-reactive cell-mediated immunity	[113]
Inactivated H1N1 influenza vaccine	Monophosphoryl lipid A (MPL) polyriboinosinic polyribocytidylic acid (poly I:C)	TLR4 TLR3	-	Increased Ag-specific antibody production and enhanced memory T and B cell responses in mice model	[115]
H1N1 influenza vaccine adjuvanted with imiquimod	Imiquimod	TLR7	-	Induced virus-specific IgM, IgG, and neutralizing antibodies	[116]
H5N1 HA vaccine adjuvanted with 3M-052	Imidazoquinoline (3M-052)	TLR7/8	-	Broadened antibody response in mice and ferrets models	[117]
rNP vaccine candidate adjuvanted with BPPcysMPEG	BPPcysMPEG (TLR2/6 agonist)	TLR2/6	-	Enhanced antigen-specific humoral and cellular responses	[127]

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
