# Peer review of "Recent Insights into the Molecular Mechanisms of the Toll-like Receptor Response to Influenza Virus Infection"

_ijms, 2024, doi:10.3390/ijms25115909_

Round 1

Reviewer 1 Report

Comments and Suggestions for Authors

The manuscript by Kayesh et al entitled “Recent Insights into the Molecular Mechanisms of TLR Response to Influenza Virus Infection” cites 120 references and reports in a balanced way the double-edged sword regarding TLR activation during influenza infection. However, certain statements are not correct and the authors need to update the review accordingly.

Major comments:

1.       Line 245 “The use of TLR agonists as vaccine adjuvants is a relatively new field…”

The discovery of for example CpG was done by Coley and the field of adjuvants targeting TLRs started many years ago.

Wiki gives “Since 1893, it has been recognized that Coley's toxin, a mixture of bacterial cell lysate, has immunostimulatory properties that could reduce the progression of some carcinomas,[4] but it was not until 1983 that Tokunaga et al. specifically identified bacterial DNA as the underlying component of the lysate that elicited the response.[5] Then, in 1995 Krieg et al. demonstrated that the CpG motif within bacterial DNA was responsible for the immunostimulatory effects and developed synthetic CpG ODN.[6] 

1.      Weiner GJ, Liu HM, Wooldridge JE, Dahle CE, Krieg AM (September 1997). "Immunostimulatory oligodeoxynucleotides containing the CpG motif are effective as immune adjuvants in tumor antigen immunization". Proceedings of the National Academy of Sciences of the United States of America. 94 (20): 10833–7. Bibcode:1997PNAS...9410833Wdoi:10.1073/pnas.94.20.10833PMC 23500PMID 9380720.

2.     ^ Bauer S, Wagner H (2002). "Bacterial CpG-DNA licenses TLR9". Toll-Like Receptor Family Members and Their Ligands. Current Topics in Microbiology and Immunology. Vol. 270. pp. 145–54. doi:10.1007/978-3-642-59430-4_9ISBN 978-3-642-63975-3PMID 12467249.

3.     ^ Rothenfusser S, Tuma E, Endres S, Hartmann G (December 2002). "Plasmacytoid dendritic cells: the key to CpG". Human Immunology. 63 (12): 1111–9. doi:10.1016/S0198-8859(02)00749-8PMID 12480254.

4.     ^ Coley WB (January 1991). "The treatment of malignant tumors by repeated inoculations of erysipelas. With a report of ten original cases. 1893". Clinical Orthopaedics and Related Research (262): 3–11. doi:10.1097/00003086-199101000-00002PMID 1984929.

5.     ^ Tokunaga T, Yamamoto H, Shimada S, Abe H, Fukuda T, Fujisawa Y, Furutani Y, Yano O, Kataoka T, Sudo T (April 1984). "Antitumor activity of deoxyribonucleic acid fraction from Mycobacterium bovis BCG. I. Isolation, physicochemical characterization, and antitumor activity". Journal of the National Cancer Institute. 72 (4): 955–62. doi:10.1093/jnci/72.4.955PMID 6200641.

6.     Jump up to:a b Krieg AM, Yi AK, Matson S, Waldschmidt TJ, Bishop GA, Teasdale R, Koretzky GA, Klinman DM (April 1995). "CpG motifs in bacterial DNA trigger direct B-cell activation". Nature. 374 (6522): 546–9. Bibcode:1995Natur.374..546Kdoi:10.1038/374546a0PMID 7700380S2CID 4261304

Hence, the knowledge that TLR agonists can be adjuvants goes back to the time when the TLRs were discovered. The authors need to update the text to more accurately reflect the history. Read additional reviews on the history of TLR adjuvants and here you need to go outside the field of influenza.^

2.       Line 321…”till date, there are no licensed TLR agonist-adjuvanted influenza vaccines”. In a review, it would be appropriate to include references/reviews from the companies making the licensed influenza vaccine to back up this strong statement. One to start with could for example be; Correlates of adjuvanticity: A review on adjuvants in licensed vaccines Giuseppe Del Giudice, Rino Rappuoli , Arnaud M. Didierlaurent, which is from 2018.

Make sure to cite information that covers all licensed vaccines as this statement is very strong. Some of the adjuvants were originally not developed as TLR-adjuvants and their TLR-adjuvant properties were discovered late in development or after licensing.

3.       In a review that highlights prospects for future influenza vaccine development and adjuvants it would be appropriate to include current problems in influenza vaccine design and the forefront, which concerns the development of a universal influenza vaccine able to mount immunity towards conserved regions to cover a large spectrum of influenza strains. Please check ClinicalTrials.gov for the latest updates.

4.       As mRNA vaccines are being developed as potential future universal influenza vaccines and the Nobel prize was rewarded based on fine-tuning the TLR activation by modifying the nucleosides it would be appropriate to reference to mRNA vaccines (Immunity, Vol. 23, 165–175, August, 2005, DOI 10.1016/j.immuni.2005.06.008).

Minor comments;

-the introduction can be effectively shorter and polished in terms of language.

-Line 39 rephrase “and so on”.

-Line 75 add CpG-DNA

-Line 74 the recognition of ssRNA by TLR7 and TLR8 is more complex-please update.

-There is repetition in the text regarding the differential expression of certain TLRs Lines 121, 133, 160- make a coherent summary of potential discrepancies instead of repeating the message three times.

-Line 174 Please spell out whether the SNPs are gain or loss of function to facilitate for the reader

-double-check that the correct writing of Proteins (TLR4) vs mRNA gene expression (Tlr4) data is being correctly referenced. Example  Sox4 line 230.

-update to include more recent studies on the efficacy of influenza vaccines. A rapid search in Pubmed gave for example  Front Immunol. 2021; 12: 744774. Published online 2021 Oct 6. doi: 10.3389/fimmu.2021.744774.

Comments on the Quality of English Language

see above

Author Response

The manuscript by Kayesh et al entitled “Recent Insights into the Molecular Mechanisms of TLR Response to Influenza Virus Infection” cites 120 references and reports in a balanced way the double-edged sword regarding TLR activation during influenza infection. However, certain statements are not correct and the authors need to update the review accordingly.

>Thank you very much for your critical comments. We have responded, as follows.

Major comments:

  1. Line 245 “The use of TLR agonists as vaccine adjuvants is a relatively new field…”

The discovery of for example CpG was done by Coley and the field of adjuvants targeting TLRs started many years ago.

Wiki gives “Since 1893, it has been recognized that Coley's toxin, a mixture of bacterial cell lysate, has immunostimulatory properties that could reduce the progression of some carcinomas,[4] but it was not until 1983 that Tokunaga et al. specifically identified bacterial DNA as the underlying component of the lysate that elicited the response.[5] Then, in 1995 Krieg et al. demonstrated that the CpG motif within bacterial DNA was responsible for the immunostimulatory effects and developed synthetic CpG ODN.[6] 

  1. Weiner GJ, Liu HM, Wooldridge JE, Dahle CE, Krieg AM (September 1997). "Immunostimulatory oligodeoxynucleotides containing the CpG motif are effective as immune adjuvants in tumor antigen immunization". Proceedings of the National Academy of Sciences of the United States of America. 94 (20): 10833–7. Bibcode:1997PNAS...9410833Wdoi:10.1073/pnas.94.20.10833PMC 23500PMID 9380720.
  2. ^Bauer S, Wagner H (2002). "Bacterial CpG-DNA licenses TLR9". Toll-Like Receptor Family Members and Their Ligands. Current Topics in Microbiology and Immunology. Vol. 270. pp. 145–54. doi:10.1007/978-3-642-59430-4_9ISBN 978-3-642-63975-3PMID 12467249.
  3. ^Rothenfusser S, Tuma E, Endres S, Hartmann G (December 2002). "Plasmacytoid dendritic cells: the key to CpG". Human Immunology. 63 (12): 1111–9. doi:10.1016/S0198-8859(02)00749-8PMID 12480254.
  4. ^Coley WB (January 1991). "The treatment of malignant tumors by repeated inoculations of erysipelas. With a report of ten original cases. 1893". Clinical Orthopaedics and Related Research (262): 3–11. doi:10.1097/00003086-199101000-00002PMID 1984929.
  5. ^Tokunaga T, Yamamoto H, Shimada S, Abe H, Fukuda T, Fujisawa Y, Furutani Y, Yano O, Kataoka T, Sudo T (April 1984). "Antitumor activity of deoxyribonucleic acid fraction from Mycobacterium bovis BCG. I. Isolation, physicochemical characterization, and antitumor activity". Journal of the National Cancer Institute. 72 (4): 955–62. doi:10.1093/jnci/72.4.955PMID 6200641.
  6. ^ Jump up to:ab Krieg AM, Yi AK, Matson S, Waldschmidt TJ, Bishop GA, Teasdale R, Koretzky GA, Klinman DM (April 1995). "CpG motifs in bacterial DNA trigger direct B-cell activation". Nature. 374 (6522): 546–9. Bibcode:1995Natur.374..546Kdoi:10.1038/374546a0PMID 7700380S2CID 4261304

Hence, the knowledge that TLR agonists can be adjuvants goes back to the time when the TLRs were discovered. The authors need to update the text to more accurately reflect the history. Read additional reviews on the history of TLR adjuvants and here you need to go outside the field of influenza.^

 Response: We are very grateful to the reviewer for pointing out this issue. In line with reviewer comments we have updated the text (line 317-322) and included the relevant references (ref. no. 97-101).

  1. Line 321…”till date, there are no licensed TLR agonist-adjuvanted influenza vaccines”. In a review, it would be appropriate to include references/reviews from the companies making the licensed influenza vaccine to back up this strong statement. One to start with could for example be; Correlates of adjuvanticity: A review on adjuvants in licensed vaccines Giuseppe Del Giudice, Rino Rappuoli , Arnaud M. Didierlaurent, which is from 2018. 

Make sure to cite information that covers all licensed vaccines as this statement is very strong. Some of the adjuvants were originally not developed as TLR-adjuvants and their TLR-adjuvant properties were discovered late in development or after licensing. 

 Response: We are grateful to the reviewer for careful reading of the manuscript and for the comments. Accordingly, we have updated the text (line 444). 

  1. In a review that highlights prospects for future influenza vaccine development and adjuvants it would be appropriate to include current problems in influenza vaccine design and the forefront, which concerns the development of a universal influenza vaccine able to mount immunity towards conserved regions to cover a large spectrum of influenza strains. Please check ClinicalTrials.gov for the latest updates.

 Response: In line with the reviewer comments, we have updated the text including the suggested points (line 339-341; 426-442; newly added Table 1).

  1. As mRNA vaccines are being developed as potential future universal influenza vaccines and the Nobel prize was rewarded based on fine-tuning the TLR activation by modifying the nucleosides it would be appropriate to reference to mRNA vaccines (Immunity, Vol. 23, 165–175, August, 2005, DOI 10.1016/j.immuni.2005.06.008).

 Response: We thank the reviewer for the nice comments. Accordingly, we have updated the text including mRNA vaccines and cited accordingly (line 426-442).

Minor comments;

-the introduction can be effectively shorter and polished in terms of language.

Response: Thank you for the comments. In line with the reviewer comments, we have shortened it and revised language (line 66).

-Line 39 rephrase “and so on”.

Response: Thank you. Rephrasing has been done (line 38-39).

-Line 75 add CpG-DNA

Response: Thank you, we have added it (line 73).

-Line 74 the recognition of ssRNA by TLR7 and TLR8 is more complex-please update.

Response: In response to reviewer comment, we have updated the text (line 73-76).

-There is repetition in the text regarding the differential expression of certain TLRs Lines 121, 133, 160- make a coherent summary of potential discrepancies instead of repeating the message three times.

Response: We have modified the text to avoid any repetition and to improve the readability (line 158-159; 169-170; 202-203).

-Line 174 Please spell out whether the SNPs are gain or loss of function to facilitate for the reader

Response: We thank the reviewer, and we have spelled the effect of SNP on TLR3 function (line 215).

-double-check that the correct writing of Proteins (TLR4) vs mRNA gene expression (Tlr4) data is being correctly referenced. Example  Sox4 line 230. 

Response: Thank you, we have made the correction (line 288).

-update to include more recent studies on the efficacy of influenza vaccines. A rapid search in Pubmed gave for example  Front Immunol. 2021; 12: 744774. Published online 2021 Oct 6. doi: 10.3389/fimmu.2021.744774.

Response: Thank you for your suggestion, we have included this recent study (ref. 109).

Reviewer 2 Report

Comments and Suggestions for Authors

This review from Kayesh et al summarizes current findings on influenza A (IAV) modulation of TLR signaling and their subsequent potential as vaccine adjuvants.  Topics covered included the contributions of different TLRs to IAV response, their contributions to IAV-onset pathology, mechanisms of IAV-initiated immune evasion (including from TLRs) and existing evidence for TLRs as use for vaccine adjuvants.  The information covered is relevant for identifying and evaluating current and future anti-IAV therapeutics.

I have the following minor comments to be addressed prior to acceptance for publication:

1) The authors should add an additional figure summarizing section 5 (summarizing IAV-mediated immune evasion) and a table summarizing 6 (to list preclinical and clinical trials using TLRs or TLR agonists as vaccine adjuvants for IAV).  In figure 1, the authors should also highlight (or otherwise indicate) the TLRs that are most significantly impacted during IAV infection.

2) Context is needed (animal or human studies) in lines 133-135 and 302-306.  The authors should also clarify the conditions in lines 159-162 that were associated with higher risk of bacterial infection and the timepoint that Pam2Cys was given before or during IAV infection in lines 193-195.

3)  In lines 169-170, the authors should clarify whether the LD50 of IAV was higher in TLR3 KO mice than WT mice, or whether they were more resistant to having 100% lethality from lethal doses in WT mice.

4) Related to lines 168-179, the authors should comment on any differences between human and mouse TLR3 signaling that may contribute to differences seen in the referenced mouse and human studies on TLR3 being pathogenic vs anti-viral.

5) The authors should comment on whether TLR agonists might mostly benefit immunocompromised individuals or whether they might also have benefits for immunocompetent individuals in overcoming current shortfalls of influenza vaccination such as strain mismatch.

Comments on the Quality of English Language

A few editing corrections for grammar needed

Author Response

Reviewer 2

This review from Kayesh et al summarizes current findings on influenza A (IAV) modulation of TLR signaling and their subsequent potential as vaccine adjuvants.  Topics covered included the contributions of different TLRs to IAV response, their contributions to IAV-onset pathology, mechanisms of IAV-initiated immune evasion (including from TLRs) and existing evidence for TLRs as use for vaccine adjuvants.  The information covered is relevant for identifying and evaluating current and future anti-IAV therapeutics.

I have the following minor comments to be addressed prior to acceptance for publication:

1) The authors should add an additional figure summarizing section 5 (summarizing IAV-mediated immune evasion) and a table summarizing 6 (to list preclinical and clinical trials using TLRs or TLR agonists as vaccine adjuvants for IAV).  In figure 1, the authors should also highlight (or otherwise indicate) the TLRs that are most significantly impacted during IAV infection.

Response: In line with reviewer comments, we have added a new figure (Figure 2) on IAV-mediated immune evasion. We also made a table summarizing the section TLR agonists as vaccine adjuvants for IAV (newly added Table 1). Also, we have stated the information of the TLRs that are most significantly impacted during IAV infection (188-190).

2) Context is needed (animal or human studies) in lines 133-135 and 302-306.  The authors should also clarify the conditions in lines 159-162 that were associated with higher risk of bacterial infection and the timepoint that Pam2Cys was given before or during IAV infection in lines 193-195.

Response: We are very grateful to the reviewer for careful reading of the manuscript and important comments. Accordingly, we have added context in the text (line 169, 385). We also have clarified the conditions as suggested by the reviewer (line 202-203, and 252). 

3)  In lines 169-170, the authors should clarify whether the LD50 of IAV was higher in TLR3 KO mice than WT mice, or whether they were more resistant to having 100% lethality from lethal doses in WT mice.

Response: In response to reviewer comment, we have updated the text (line 211-212).

4) Related to lines 168-179, the authors should comment on any differences between human and mouse TLR3 signaling that may contribute to differences seen in the referenced mouse and human studies on TLR3 being pathogenic vs anti-viral.

Response: We thank reviewer. We have accordingly updated the text (line 220-222).

5) The authors should comment on whether TLR agonists might mostly benefit immunocompromised individuals or whether they might also have benefits for immunocompetent individuals in overcoming current shortfalls of influenza vaccination such as strain mismatch.

Response: In response to reviewer comments, we have added the benefits of TLR agonist use in vaccine (line 423-425; 448-450).

Round 2

Reviewer 1 Report

Comments and Suggestions for Authors

Thank you for the updated version. No further comments.

Comments on the Quality of English Language

Minor misspellings found in the new text that can be corrected during proof-reading.